# Perspectives of Immigrants and Native Dutch on Antibiotic Use: A Qualitative Study

**DOI:** 10.3390/antibiotics11091179

**Published:** 2022-08-31

**Authors:** Dominique L. A. Lescure, Alike W. van der Velden, Natascha Huijser van Reenen, Jan Hendrik Richardus, Helene A. C. M. Voeten

**Affiliations:** 1Department of Public Health, Erasmus MC, University Medical Center Rotterdam, Doctor Molewaterplein 40, 3015 GD Rotterdam, The Netherlands; 2Department of Infectious Disease Control, Municipal Public Health Service Rotterdam-Rijnmond, Wilhelminakade 139, 3072 AP Rotterdam, The Netherlands; 3Julius Center for Health Sciences and Primary Care, University Medical Center Utrecht, Universiteitsweg 100, 3584 CX Utrecht, The Netherlands; 4Pharos (Dutch Centre of Expertise on Health Disparities), Arthur van Schendelstraat 600, 3511 MJ Utrecht, The Netherlands

**Keywords:** emigrants and immigrants, anti-bacterial agents, communication, primary health care, qualitative research, health knowledge, attitudes, practice

## Abstract

Immigrants constitute large proportions of the population in many high-income countries. Knowledge about their perceptions of antibiotics, in comparison to native populations, is limited. We explored these perceptions by organizing nine homogeneous focus group discussions (FGDs) with first-generation immigrant and native Dutch participants (N = 64) from Rotterdam and Utrecht, who were recruited with the assistance of immigrant (community support) organizations. The FGDs were audio-recorded and transcribed verbatim. Inductive thematic analyses were performed with the qualitative analysis software Atlas.ti, using open and axial coding. We did not find noteworthy differences between immigrants and native Dutch participants; all participants had an overall reluctant attitude towards antibiotics. Within-group differences were larger than between-group differences. In each FGD there were, for instance, participants who adopted an assertive stance in order to receive antibiotics, who had low antibiotic-related knowledge, or who used antibiotics incorrectly. Native Dutch participants expressed similar difficulties as immigrant participants in the communication with their GP, which mainly related to time constraints. Immigrants who encountered language barriers experienced even greater communicational difficulties and reported that they often feel embarrassed and refrain from asking questions. To stimulate more prudent use of antibiotics, more attention is needed for supportive multilingual patient materials. In addition, GPs need to adjust their information, guidance, and communication for the individual’s needs, regardless of the patient’s migration background.

## 1. Introduction

In various countries, antibiotics are often sold over the counter without the need of a medical prescription [1]. This unregulated sale of antibiotics induces inappropriate use [2]. Non-prescription antibiotic use is especially common in low- and middle-income countries, with a range from 8% to 93% [3,4,5,6]. In addition, antibiotics are often prescribed unnecessarily in primary care by general practitioners (GPs); for example, in Poland, 50% of patients receive antibiotics for a common cold, and in Canada, the unnecessary prescribing rates are highest for acute bronchitis (52.6%) [7,8]. The overuse of antibiotics not only leads to unnecessary side effects, such as rash or diarrhea, but also to the emergence of antibiotic resistance [9,10].

Immigrants may have healthcare perceptions that differ from the native population regarding antibiotic use. These perceptions are influenced by one’s migration experience and by cultural values and attitudes that are shaped by contrasting socialization processes in various healthcare systems [11,12]. Recent studies showed that Turkish and Syrian immigrants in Europe have higher consumption rates of antibiotics than native populations, share antibiotics among family members, and buy antibiotics without prescriptions in their home country [13,14]. Additionally, for immigrants in New Zealand and Australia, it was found that they are more likely than the native population to import antibiotics from their home country or keep leftovers for future use [15,16]. The higher consumption rate among immigrants is reflected in experiences of GPs, who feel that immigrants implicitly and explicitly request antibiotic prescriptions more often than the native population [13]. Evidence also shows that immigrants have different perspectives and knowledge regarding antibiotic use. For instance, immigrants reason that other medications will be less effective for them because they are ‘used to’ antibiotics [17], they strongly believe in the curative power of antibiotics [14], and they have a lower level of antibiotic knowledge [18].

Immigrants may encounter additional barriers resulting from poor language proficiency and from the inability of healthcare institutions to meet their cultural needs [19]. The presence of a language barrier can lead to a higher chance of receiving antibiotics [20], which indicates that effective communication is crucial for the appropriate use of antibiotics. Previous studies have described that immigrant patients did not receive adequate information on how to use antibiotics appropriately and suffered from a lack of suitable information materials, especially in their mother tongue [13,21]. According to immigrants themselves, communication with their healthcare provider is important for the appropriate use of antibiotics, which underlines the urge to improve GP–patient communication [21].

The Netherlands is a country with a growing population of immigrants who originate from all parts of the world. Many of these immigrants arrived from the mid-1960s onwards as ‘unqualified guest workers’. Furthermore, a large part of immigrants are refugees that originate from former Dutch colonies, or arrived because of family reunification. Currently, 25.8% of Dutch inhabitants have a migrant background (first and second generation) [22]. Because of their familiarity with healthcare systems with less strict antibiotic guidelines, it can be expected that immigrants hold antibiotic perceptions that differ from the native Dutch population. Although the Netherlands is, within Europe, one of the countries with the lowest antibiotic consumption rates due to its strict antibiotic policies [23], inappropriate prescription is still substantial; for example, 46% of the prescribed antibiotics for respiratory tract indications (RTIs) are not in accordance with Dutch guidelines [24]. A considerable number of Dutch interventions have been developed, aimed at reducing antibiotic prescriptions and at improving patients’ knowledge about antibiotic use [25,26]. Most of these interventions pay no attention to different population groups, while it can be expected that many immigrants have difficulties with adapting to the strict Dutch antibiotic policies. Few studies have focused on the perceptions of immigrants towards antibiotic use in relation to the prescription behavior of GPs [14,21].

To encourage more appropriate antibiotic use and to rationalize antibiotic prescriptions, it is important to have knowledge of the perceptions of different population groups. These insights can help improve both the antibiotic prescribing behaviors of GPs and GP–patient communication, as well as help optimize adherence [27,28]. Our study explores the perspectives of different population groups on antibiotic use and the related GP–patient communication. This study is performed as part of the PARCA-project (Prescription of Antibiotics in pRimary Care; a focus on immigrant communities), which aims to stimulate appropriate prescribing of antibiotics by GPs.

## 2. Materials and Methods

### 2.1. Study Design

We used a basic qualitative descriptive research design and conducted focus group discussions (FGDs).

### 2.2. Setting, Participants, and Sampling

We used a purposeful sampling technique to recruit men and women, aged 18+, from different population groups. We conducted FGDs until overall data saturation was attainted about antibiotic use, meaning that we did not strive for data saturation within the various population groups from all countries, but within the total of the FGDs.

We recruited participants who were originally from Turkey, Morocco, or Surinam. These immigrants represent the three largest immigrant groups in the Netherlands [22]. In addition, we recruited Syrian immigrants, as they constituted the largest group of asylum seekers [22]. Furthermore, we focused on the recruitment of Cape Verdean immigrants because they are one of the top five ethnic groups in Rotterdam [29]. Ethnically homogeneous FGDs were selected because they were expected to encourage participants to discuss topics in a comfortable environment [30]. For the recruitment of the immigrant participants, we collaborated with immigrant (community support) organizations in Rotterdam and Utrecht. These volunteer-based organizations serve immigrants from different ethnic groups by offering programs and weekly activities like health education, child-raising support, and cultural events. For each immigrant organization, we contacted the agency coordinator to assist in recruitment. We included first-generation immigrants with differing educational levels who had used antibiotics at least once. In close contact with the agency coordinator, and depending on the activities of the specific organization, we decided whether we wanted to include both men and women or to organize a single-sex group to ensure each participant would feel secure to speak out freely. Subsequently, the agency coordinators recruited participants actively by announcing the study during their weekly activities. This recruitment method guaranteed the privacy of the participants.

Besides immigrant participants, we recruited native Dutch participants with differing educational levels, for comparison reasons. These participants were recruited through the Digital City Panel Rotterdam, which consists of about 7000 Rotterdam inhabitants. Panel members are approached several times a year to participate in short surveys on a broad range of subjects, which helps improve the local policy of the Municipality of Rotterdam. We invited all members of the Digital City Panel to indicate their interest in joining a FGD about antibiotic use. We also recruited Dutch participants with limited literacy skills through the agency coordinator of ‘Stichting ABC’, which is a volunteer-based organization for people with limited literacy.

### 2.3. Data Gathering

To facilitate the FGDs, we used a topic-guide with semi-structured questions. These were based on the literature and focused on experiences with (1) the use of antibiotics, and (2) (communication with) the GP in the Netherlands. Prior to the FGDs, the topic guide was reviewed by health professionals and health literacy experts and was modified based on their comments. All FGDs were in Dutch. As we needed to ensure the questions in the topic-guide were also understandable for participants with limited language abilities and/or limited literacy skills, we formulated a limited number of simple, straightforward, open questions. We used the same topic guide for all FGDs.

The FGDs were moderated by either DL, a sociologist with experience in qualitative research, or by NHvR, who has a master’s degree in medicine. The times and venues of the FGDs with immigrants and participants with limited literacy were decided in consultation with the agency coordinators. The native Dutch FGDs with members of the Digital City Panel took place at a municipal building in Rotterdam. FGDs were scheduled to last 90 min.

At the start of each FGD, the moderator briefly introduced herself, described the purpose of the study, and read an informed consent statement, to which the participants orally consented. All participants were informed in advance about the study, and they had the opportunity to ask questions. No prior relationships existed between the moderators and the participants. All FGDs were audio-recorded with the participants’ consent. At the end of each FGD, the moderator and the note-taker debriefed about the key findings and identified any need for further clarification. To acknowledge the time they had invested, participants received a twenty-euro gift card upon the conclusion of each FGD.

### 2.4. Data Analysis

The recordings of each FGD were transcribed verbatim by an independent professional transcription company. References to individuals were removed to anonymize the transcripts. We used the qualitative software program Atlas.ti to analyze the data through inductive thematic analysis. First, by reading and re-reading the transcripts, DL used open coding to attach labels to quotes. An independent research assistant with a medical background double-coded one third of the transcripts to enhance the reliability and validity of the analyses. Codes were compared and discussed to create a codebook. Existing discrepancies were discussed until consensus was reached. DL expanded or merged themes to refine the codebook. HV reviewed the codebook to discuss the identified themes and sub-themes and to reach a final agreement with DL. Subsequently, DL used axial coding to search for relationships between open codes and to search for central themes. Themes and sub-themes were then identified and examined across all FGDs. We complied with the COREQ (consolidated criteria for reporting qualitative research) checklist in reporting this study [31].

### 2.5. Ethical Considerations

Ethical approval for this study was waived by the Medical Ethics Review Committee at Erasmus MC, University Medical Centre Rotterdam (MEC-2018-1628) because this was not a medical–scientific experiment on human subjects. For the FGDs with immigrants and the participants with limited literacy, the research team wrote a letter about the aim and content of the FGD in simple wording. This letter was handed over by the agency coordinators to the participants and discussed prior to the start of each FGD. We used the same information letter for the other native Dutch participants, but sent it by email before the FGD. All participants were assured about their anonymity and confidentiality. Furthermore, they were assured that participation was voluntary, and that they could leave the session at any time. The transcripts and the audio-records were anonymized. They were kept secure in accordance with the guidelines of the Erasmus MC.

## 3. Results

We conducted nine FGDs with 64 participants between January and November 2019. The FGDs lasted between 85 min and 2 h. If a participant had difficulties with understanding the questions or with expressing him-/herself in Dutch, one of the other participants acted as translator. In the Syrian group, the conversation was partly in English because two of the participants sometimes felt most comfortable in English.

Table 1 provides a detailed overview of participants. We organized one FGD with all immigrant groups, except for Turkish immigrants, for whom we organized two FGDs to facilitate the large number of participants. We also aimed to organize a second FGD with Syrian participants, but unfortunately this turned out to be unfeasible. The FGDs were organized either in Rotterdam or in Utrecht. The sample of the immigrant participants comprised 35 women and 11 men and represented a broad range of ages. More than half of the participants were lower educated (63%), and more than half had been living in the Netherlands for longer than thirty years (56%). We organized three FGDs with native Dutch participants. Males and females were equally represented. More than half of the participants were lower educated (56%) and older than sixty (61%).

In the thematic analysis, the following six main themes emerged: (1) Reluctant antibiotic attitude; (2) Wide variety in antibiotic knowledge; (3) Adopting an assertive stance in order to receive antibiotics; (4) Inappropriate antibiotic use; (5) Language barriers; (6) Information provision about antibiotics.

Most of the main themes were further divided into subthemes. The data regarding the themes were compared across the immigrant and native Dutch groups. In general, we did not find noteworthy differences between immigrant and native Dutch participants. In all FGDs, there was considerable variation between participants in how they perceived the discussed themes, and none of the arguments presented were restricted to one specific population group. Moreover, the native Dutch participants experienced the same hurdles while receiving antibiotic care as the immigrant participants did. As a result, almost all themes and subthemes are discussed for all participants combined. Theme 5, ‘language barriers’, was the main exception as this theme was only discussed in the FGDs with immigrant participants.

The qualitative data were summarized and are presented below with illustrative participants quotations.

## 4. A Reluctant Antibiotic Attitude

### 4.1. Reluctance Determined by Experience and Knowledge

All participants mentioned only wanting antibiotics when there were no other treatment options. Most of them reported trying to first manage their symptoms at home without medical care: *“No, I almost never use antibiotics. The same goes for my children. If I am ill, I take a rest and eat a healthy diet. Then, if the illness is not severe, the symptoms will disappear”* (Turkish woman). There were participants with either a very positive or a very negative experience with antibiotic use. Participants with a positive experience underlined that the antibiotics were highly effective: *“I did not have side effects and I recovered really fast. I am actually a fan.”* (native Dutch). Another part of the participants had a negative experience with the use of antibiotics, mainly because of severe side effects. These participants mentioned the side effects as a decisive reason for being reluctant. Some participants were reluctant with the use of antibiotics because of the risk of antibiotic resistance. Concerns regarding antibiotic resistance were generally related to risks for participants’ own health, except for two participants who also mentioned the public benefit: *“If you are ill, you want to recover as quick as possible, but sometimes it is better, I think, for the general good, to be ill for another week, instead of using immediately a heavy antibiotic course […] It is not only for yourself, because bacteria in general are becoming resistant against antibiotics, so […] there is an increase in super bacteria for which we only have limited effective medicines.”* (Surinamese man). A small part of the participants described the broad impact of antibiotics on the body, as healthy bacteria can also be killed. Participants feel this makes them more susceptible to other diseases and, because they were aware of this fact, they are careful regarding the use of antibiotics.

### 4.2. Increased Reluctance of Immigrants

In general, the immigrant participants stated that they trust the decision of their Dutch GP and accept the situation in which they receive an alternative treatment proposal instead of antibiotics. Some participants felt reassured when they did not receive antibiotics because that indicated they had a mild illness: *“I would rather not use antibiotics. It [not receiving antibiotics] is good news because it is based on the judgement of the GP about the severity of the symptoms. […] If the GP says, ‘You don’t need antibiotics’, I am happy”* (Cape Verdean woman). All immigrant participants were used to a high consumption and easy accessibility of antibiotics in their country of origin: *“The difference with Syria is that we can buy it at the pharmacy. As a result, people buy it often.”* (Syrian woman). A large number of the immigrant participants explained that their perspective on antibiotic use has changed since their arrival in the Netherlands. They have become more reluctant because they have learned from the GP that their own body has the ability to recover by itself: *“When we are sick in Turkey, we go to the doctor. He always gives antibiotics. We believe antibiotics are always good. When I came in the Netherlands, I became sick. And then I went to the GP. She did not give me antibiotics and told me to rest and lie down and that the disease would disappear. Yes, then I was angry. But now I understand antibiotics are not good, I never use them.”* (Turkish woman). Immigrant participants mentioned having learned antibiotics are not always the best solution because of their potential side effects: *“Yes, currently, all mothers know exactly that antibiotics are not normal medicines, that they are not paracetamol. They also have disadvantages, and it is not good if they are always accessible.”* (Syrian woman). None of the immigrant participants stated that they buy antibiotics in their country of origin.

### 4.3. Increased Prudence of Dutch GPs

Among the native Dutch participants, it was a general opinion that Dutch GPs have become more prudent to prescribe antibiotics: *“In fact, it is something of the last four years that GPs are very reluctant. And that is the reason why the GP asked me ‘do you want antibiotics, or do you want to see if you can recover by yourself?’ So, I see a switch from prescribing as much as possible to rather not”* (Dutch man). A few of the native Dutch participants described that, several years ago, they sometimes received antibiotics for infections that are commonly caused by viruses. Currently, as participants explained, this no longer happens. According to most of the native Dutch participants, there is more focus on the human body’s ability to self-heal nowadays. Furthermore, the participants reported that, over the years, the GPs’ information provision and the involvement of the patient in decision-making has increased. This subject was not raised by the participants of the immigrant groups.

## 5. Wide Variety in Antibiotic Knowledge

There was large variability within each FGD in participants’ knowledge about antibiotics and antibiotic resistance, with knowledge varying from poor to good. Frequent users of antibiotics—mainly those with a chronic disease—displayed a higher knowledge level. These participants described, for instance, that antibiotics are only effective against bacterial infections and that antibiotics sometimes can induce sun sensitivity. Participants with a lower knowledge level thought that antibiotics are necessary in treating viral infections. Moreover, they also believed that the frequent use of antibiotics weakens the body and reduces its ability to heal: *‘The strength of our body decreases. It decreases and cannot heal anymore in itself.’* (Syrian woman). A small number of participants were convinced that antibiotics create an adverse reaction in the body and will create antibodies against the antibiotics. Several participants expressed an understanding of antibiotic resistance by using their own wordings: *‘When I stop the antibiotic treatment too quickly, bacteria in my body will become stronger and the next time these bacteria will not be defeated by the same antibiotics. So, the antibiotic will become a sweet for the bacteria. The bacteria will say, ‘come to me, I will eat you, I will not die from you’’* (Surinamese woman). A few participants mentioned that antibiotics not only fight the ‘bad’ bacteria, but can also destroy healthy gut bacteria.

## 6. Adopting an Assertive Stance to Receive Antibiotics

### 6.1. Ways of Receiving or Demanding Antibiotics

Approximately half of the participants, both immigrants and native Dutch, admitted that they sometimes exaggerated their symptoms, or those of their child, to ensure they would receive antibiotics from their GP. In all FGDs, participants were aware that you will not receive antibiotics when you are honest: *‘You have to lie to receive an [antibiotic] treatment and say your complaints are lasting longer than a week.’* (Cape Verdean woman); *‘I have learned you need to exaggerate your complaints’* (native Dutch). One participant elaborated on a single situation in which she urged the GP to supply her a delayed antibiotic prescription: *‘Then I said ‘you know what; just prescribe and I will wait 2 to 3 days to see if the symptoms will resolve or not. Only if the symptoms worsen, I will go to the pharmacy to get the antibiotic course.’ So, the GP prescribed me an antibiotic course. Immediately afterwards, I visited the pharmacy for the antibiotics because I will not wait 2 to 3 days’* (Turkish woman).

Moreover, in all FGDs, at least one participant came up with another way to receive antibiotics because of their determination that antibiotics were needed. A few of these participants stated they sometimes directly asked the GP for antibiotics: *‘Two months ago my son was very sick and could not breathe normally. And his throat had an awful smell. I kindly asked the GP ‘I want you to give my son an antibiotic’ [But the GP replied] ‘No, he has to recover by himself’* (Turkish woman). Two native Dutch participants emphasized they were stubborn during consultation with the pursuit to receive antibiotics: *‘I said to the GP ‘I sit here and will not go away until I receive antibiotics. Twenty minutes later I went outside [with antibiotics]’.* (native Dutch). *‘I needed to be persistent to see the GP and would have been sent home without antibiotics if I had not been assertive enough. So, I needed to put in extra effort because I came to the GP to receive antibiotics’* (native Dutch).

Some participants reported that they suffer from a chronic disease and, as a result, always request a preventive antibiotic course when they go on vacation abroad: *‘When I go on holiday I say to my GP ‘Listen to me, I will go on holiday outside of Europe and need an antibiotic course’ Then the GP will provide me the course’* (native Dutch). One female participant with repeated UTIs mentioned that she regularly buys antibiotics in Spain on her own initiative: *‘In Spain, you buy antibiotics at the pharmacy for four euros.’* (native Dutch). She explained that she uses the antibiotics to bridge the gap until the GP can prescribe an evidence-based antibiotic course.

### 6.2. Situations in Which Antibiotics Are Desired

Adopting an assertive stance in order to receive antibiotics was discussed in all FGDs in relation to specific situations. First, according to most of the participants, the prescribing behavior of the GP is incomprehensible and too restrictive when their child has very severe symptoms: ‘*[The GPs] are terribly strict with antibiotics. I believe they are too strict because the GP will prescribe antibiotics only after five days. I know antibiotic use is better for my children. Here, children are getting sicker.*’ (Syrian woman). Not receiving antibiotics in these situations leads to many emotions: ‘*I swear, in a situation with my son, I was furious because I did not receive antibiotics easily. You only receive antibiotics after a week, after severe worsening of [your child’s] symptoms. It isn’t easy.*’ (Turkish woman); ‘*You feel angry and desperate, and you do not know what to do. Then, once in a time, you will exaggerate a bit to receive the antibiotics you want.*’ (Turkish woman). Participants believed it is important that their children rapidly receive antibiotics, because it is unnecessary for children to build up immunity at a young age. In addition, participants reported that sometimes it was urgent that they themselves directly received antibiotics in order to ensure that they could take care of their children.

Second, a number of participants emphasized the need for antibiotics because of other obligations, either social or work related. Participants mentioned they needed antibiotics during hectic periods, for instance due to working overtime, as antibiotics can function as a quick fix. In addition, a few participants desired antibiotics because of the fear of missing out social events: ‘*It happens that you have a reunion that weekend. Then you want to be fine and want to use antibiotics. That is how it works. Sorry…*’ (native Dutch).

Third, participants desired antibiotics to relieve symptoms or pain, specifically when the pain had already lasted for a long time: “*Recently, I visited the GP because I suffered from pain in my sinuses. So, I thought ‘this might be a sinusitis’. […] I know it takes a while before the symptoms disappear. I needed my energy, so I visited the GP with the hope to receive antibiotics.*” (native Dutch). Within all FGDs, several participants claimed that their self-judgement about a possible infection and the need for antibiotics is superior to the clinical judgement of a GP. This was mainly related to those who regularly suffer from urinary tract infections (UTIs): ‘*When I call the GP and I explain him I have a urinary tract infection, he says ‘Wait, I first want to see a urine sample*.’’ I say ‘*I have it for the millionth time, so I know how it feels. I believe it was nonsense to submit a urine sample again in such a situation. I mean, I am not retarded*’ (native Dutch).

## 7. Inappropriate Antibiotic Use

Almost all participants, both immigrants and native Dutch, experienced difficulties with the appropriate use of antibiotics. In general, participants mentioned that this is due to their GP not providing enough information and using unclear terms. Participants stated that many problems are related to the dosage schedule. They perceive the intake of antibiotics at regular intervals as an almost impossible task if they also have to align this with their meals. Several participants reported that their irregular eating patterns resulted in omitted intake moments. Moreover, some participants were uncertain about the meaning of the term ‘meal’ and questioned, for instance, whether eating only one sandwich would be sufficient.

A large number of participants indicated that they had once stopped too early with the antibiotic course when their symptoms had resolved: ‘*You feel fine and after three days you just quit*’ (native Dutch). Some participants discontinued their antibiotic treatment because of the simultaneous use of other medicines: ‘*You have to finish the [antibiotic] course. But I must admit I not always adhered to that. At a certain point I thought ‘it is a medication and if I do not have to take it together with my other medications than that will be better. So, I did that often in the past*’ (native Dutch). One participant quit the antibiotic course too early because of severe side-effects, and one participant quit because she forgot to take the antibiotics with her on holiday.

Almost all participants who discontinued their antibiotic course underlined that they would not use the leftover antibiotics a next time on their own initiative, except for two participants (one native Dutch and one Turkish woman): ‘*I have used those [antibiotic pills] the next time. Not exactly with the same symptoms but with the idea ‘I will get sick again*’ (Turkish woman).

## 8. Language Barriers

### 8.1. Language Barriers Impede GP-Patient Relations

Most immigrant participants learned to communicate with their GP in a second language that they and their GP both speak, such as English. Still, almost all immigrant participants revealed that, when they had first arrived in the Netherlands, they faced significant language barriers that hindered them to effectively communicate with their Dutch GP: ‘*I was in panic. What can I do? In my language it was terrible. I could not tell anything*’ (Syrian woman). Participants observed that GPs behave differently when the patient has limited Dutch proficiency: ‘*You feel certain irritations from the GP if you do not know the Dutch language and then the GPs will behave differently. They are more likely to say, ‘We need to wait a few days.’ It feels that they try to blow you off*’ (Turkish woman).

Some of the participants continue to experience communication difficulties with their GP. As a result, they stressed that they often feel embarrassed and refrain from asking questions: ‘*Patients feel afraid to ask questions, that is a problem*’ (Moroccan man). While discussing this subject more into depth, a few participants mentioned being critical towards the proactive attitude that is required from Dutch patients, which is more difficult in language-discordant conversations: ‘*If you speak Dutch fluently you can ask additional questions and find out that you are entitled to certain things [like a specific treatment]*’ (Turkish woman). Furthermore, a considerable part of the participants who had difficulties with speaking Dutch, felt ignored by their GP: ‘*I do not have trust because no one is listening. I have had so much pain, but they are not listening*’ (Cape Verdean woman). Some of them explained this can lead to medical care avoidance: ‘*If you are treated like they do not listen to you, then the next time you are sick you will not ask for help that quickly*’ (Turkish woman). One participant reported that she visits GPs in her home country: ‘*Sometimes I do not believe the Dutch GP; therefore, I visit the doctor in Turkey. I do the same for my children. We have often an extra check in Turkey*’ (Turkish woman).

### 8.2. Participants’ Responsibility for Overcoming Language Barriers

A major complaint that immigrant participants discussed was that GPs do not provide solutions to overcome language barriers. Consequently, as the participants explained, the information provided by GPs is incomprehensible, especially when complex words are used: *‘It has to do with the medical terms. You think, what do they mean with that?’* (Surinamese man). Sometimes, GPs try to improve the communication using leaflets, but these are often only available in Dutch. Often, participants feel that the GP places the full responsibility on them to bring along an informal interpreter. Participants perceive this as a significant hurdle when they want to have a private, intimate conversation with their GP. Moreover, several participants explained it is not only uncomfortable for themselves, but it also poses a high (emotional) burden on the translator. Some participants have experienced that themselves: *‘If my friend visits the GP, she asks for an interpreter, but he never has one. So, I always join her which is also difficult for me [because of the long distance]’* (Syrian woman). These communication challenges lead to a lack of understanding the GPs’ advice and forces participants to search the internet for information on their problem in their own language on websites from their home country: *‘When I have fever or when I have pain, I search on YouTube, type my disease and listen to GPs who explain the symptoms. We have similar health websites in Turkish as you have in Dutch. They write in Turkish. But you only write in Dutch so people will search for other websites’* (Turkish woman). A few participants tended to rely on the experiences of family and underlined that their family’s advice is superior to their GP’s guidance: *‘We have an expression in Morocco: when someone has experience, it is better than the doctor’* (Moroccan man).

### 8.3. Need for Provider Solutions to Overcome Language Barriers

All participants distinguished bilingual pharmacists or GPs as a helpful solution to overcoming language barriers. Participants also valued receiving written information on medications, which can be reread at home. Some of the participants preferred to receive the information in their mother tongue, while others desired to receive the information additionally in Dutch: *‘For me, in Dutch, because if the GP explains something than I recognize it because I have heard or read about it before. It is necessary in Dutch […] because we live here and have direct contact with the GP’* (Syrian woman).

The other possible solutions that were mentioned for overcoming language barriers concerned the use of a professional telephone interpreter service, the availability of multilingual information videos on social media, and the use of pictures, such as in folders, to clarify the verbal explanation of the GP. Especially for intimate conversations, participants explicitly mentioned that independent professional translating solutions are crucial. If these solutions are not available, patients avoid care: *‘If I want to visit the GP, I am afraid to make an appointment, because I first have to find out if someone, I trust can join me. If there is no one, I will not go. That is my problem.’* (Cape Verdean woman). Furthermore, participants reported that the GP needs to facilitate sentence-by-sentence translation: *‘Many people drop out because they cannot speak Dutch and then they ask me to translate for them. But I do not get the possibility to translate between the sentences’* (Turkish woman).

## 9. Information Provision about Antibiotics

### 9.1. Importance of (Time for) Communication and Trust in GP

In all FGDs, the participants presented a broad range of opinions on the information provided by the GP. Satisfied participants highlighted that the obtained information was sufficient and understandable, and they felt that they did not want or need more information. Conversely, there were participants who were dissatisfied because of unmet information needs, such as knowing why their GP is reluctant to provide antibiotic treatment or knowing why they do not provide antibiotics: *‘I never receive information about my condition. I receive an antibiotic treatment, but they never tell for which bug it is’* (native Dutch). Participants explained that the information provided by the pharmacist is preferable, because it is more comprehensive: *‘I receive more information from my pharmacist than from my GP’* (native Dutch).

The limited available time during consultations was described as the most important barrier for sufficient information provision: *‘There is lack of time with GPs. It [the use of antibiotics] requires a lot of explanation. Many GPs do not have time for that’* (native Dutch). Furthermore, almost all participants reflected that time constraints hinder them in discussing multiple symptoms with their GP and, as a result, they complained that they felt that they were not being taking seriously. They feel that there is no attention for them as an individual, and that the GP is unable to make a correct diagnosis because physical examination is often lacking: *‘Here, [the GP] only uses his mouth, he only talks ‘where do you have pain, where do you have something…*’’ (Moroccan man). A few participants were familiar with the possibility to ask for an extended consult. This option is sometimes used to ensure that there is enough consultation time to discuss all of their questions. All immigrant participants agreed that the time constraints also limit the extent to which GPs can overcome a potential language barrier.

The majority of participants who use health information from the internet to increase their understanding of their symptoms emphasized that they experience difficulties in interpretating this internet information, its trustworthiness, and the personal relevance: *‘Then I look on YouTube, but not everything on YouTube is good. I will try things when I believe it is good. But not always the information is good’* (Moroccan man). Therefore, participants prefer spoken information from their GP because they rely on their GP for the interpretation of the internet information: *‘I do not like the impossibility to validate the information. […]. I cannot tell whether the same accounts for me. And I always want to hear from the GP what it is. […] Then you can ask something and clarify things.’* (native Dutch).

All participants in all FGDs highly valued personal contact with the GP. They explained that personal contact with the GP is crucial for building a trust-based relationship. If there is no trust, there is a risk of care avoidance: *‘The GP is the first one, the front line. I must trust him. If there is no trust, then it does more harm than good. Then I will avoid the GP. And then he wants to see me way earlier, but then I will not come. And then I will come only when it is already too late. And then he will say as a GP ‘Why did you wait so long?’ ‘Because you were not listening’’* (native Dutch).

### 9.2. Large Variation in Information Provision Preferences

In all FGDs, a discussion arose around preferred information provision, in relation to both content and form. Participants found that health information should include information about how the GP decides whether or not to prescribe antibiotics, possible alternative treatment options, and the (dis)advantages of taking antibiotics. Moreover, several participants desired more practical information about what food they can take with antibiotics, the interaction of antibiotics with other medicines, and why it is important to finish the whole course. Regarding the form of information provision, participants unanimously agreed that receiving information from the GP is paramount. Apart from that, there was a wide variety in other preferred information forms. Participants wished, for example, to be informed by apps, by a GP from their home country, short information movies on TV with subtitles, or leaflets with pictures: *‘Visual information is often very easy, you can take it with you, you go home and can reread the text. It is always useful if the GP practice has somewhere a leaflet in multiple languages’* (Cape Verdean woman). Some participants disagreed on the use of leaflets: *‘Really, no one is reading written information, especially not the elderly’* (Turkish woman). Several participants believe it is necessary to start education about antibiotic use and other medicines in primary schools. In their opinion, children need to be educated at a young age on how to prevent sickness and disease. A small minority of participants would appreciate a control consultation with their GP after ending the antibiotic course, in which they can ask questions about persisting symptoms. Finally, according to the participants, the provided information needs to be simple, practical, short, and available in multiple languages.

## 10. Discussion

We explored the perspectives of different immigrant and native Dutch groups on antibiotic use and the related GP–patient communication. We did not find noteworthy differences between immigrant and native Dutch participants. In general, participants had a reluctant antibiotic attitude. They mentioned multiple prominent problems, such as time constraints during consultations or difficulties with understanding medical wordings used by the GP. These can be considered as universally experienced problems that were unrelated to one’s ethnic background and/or specific topic. There was also a universal preference for receiving information personally from the GP. Among all immigrant groups, we found that being hindered during consultation with the GP due to a language barrier was a crucial theme and was underlined by all participants. For the other themes, we found that within-group differences were larger than between-group differences. For example, each immigrant and native Dutch group contained individuals who adopt an assertive stance in order to receive antibiotics and people who do not, as well as people who incorrectly use antibiotics and people who use antibiotics correctly. This might indicate that personal characteristics other than ethnic background are decisive in explaining someone’s attitude towards antibiotics.

The largest groups of immigrants, Turkish, Moroccan, and Surinamese immigrants, arrived in the Netherlands between 1960 and 1980. This is reflected in the age distribution of these groups: around 60–70% of them are older than 45 [22]. The same holds for the immigrant participants of our study, as most of them have been living in the Netherlands for more than 10 years (80%) and are older than 45 (>54%). As a result, these participants have had a long time to adapt to Dutch antibiotic guidelines and Dutch routine GP practice. This might explain why we found considerable similarities between immigrant and native Dutch participants regarding antibiotic attitude and use. In the literature, discussions often focus on differences between first- or second-generation immigrants [13,18], but it can be expected that there are also differences between first-generation immigrants depending on their length of stay in the host country. Nevertheless, similar findings were also reported by other recent studies. The study of Alkirawan (2022) [14] shows that Syrian immigrants, after arriving in the Netherlands, accepted the advice of Dutch GPs when they did not prescribe antibiotics and that they became more aware of the negative effects of antibiotic use. In addition, two other recent studies found that Turkish immigrants in Europe seldom use antibiotics without a prescription and express similar attitudes and expectations as the native population [13,21]. These studies also show that most immigrant patients had good knowledge about antibiotics, did not express many questions when antibiotics were prescribed, and that their requests for antibiotic prescriptions are not considerable and have declined in recent years [13,21].

Both of these recent studies, as well as our own qualitative study, underline the worldwide trend towards lower expectations to receive antibiotics [32] and a decreasing trend in total antibiotic use in the European Union overall between 2011–2019 [23]. Moreover, they are consistent with the study of Schuts et al (2019) [18], which demonstrated no differences between six Dutch ethnic groups in the number of antibiotics used. Besides adapting to antibiotic guidelines in their host country, the reluctant antibiotic attitude of immigrants can also be derived from the implementation of antibiotic stewardship programs such as the National Action Plan (NAP), which was introduced in Turkey in 2014 [21,33]. The NAP resulted in a reduction of antibiotic prescriptions from 34.9% in 2011 to 24.6% in 2018 [34]. Moreover, Turkish inhabitants nowadays show appropriate attitudes and practices towards antibiotics and only a small portion declared that they put pressure on healthcare professionals to prescribe antibiotics [33]. Not all recent Turkish studies are predominantly positive about Turkish people’s attitudes and knowledge [35,36] and show that 16.9% of patients still force the physician to prescribe antibiotics [37], which may underline strong individual differences. Another explanation for the similarities between immigrant and native Dutch participants might be that cultural differences in healthcare mainly play a role when there are psychological issues or traumatic experiences that involve complex treatments [38,39].

For the FGDs, we succeeded in including both lower- and medium/higher-educated participants in all groups (63% were lower-educated in the immigrant group and 56% in the native Dutch group). Surprisingly, we did not find very clear differences between these lower- versus medium/higher-educated participants. Among the higher-educated participants, there were also individuals who desired more information from their GP, put pressure on their GP, have difficulties with following strict dosage schedules, or who are uncertain about specific terms and their meanings. Other studies showed that people who have a lower socioeconomic status or education level have lower knowledge of antibiotics, have higher expectations for antibiotics to be prescribed, have a higher chance of misusing antibiotics, and are more likely to demand antibiotics for a cold or flu [40,41,42]. Although this seems to contrast our results, it might indicate that medium/higher-educated participants wrongly use or expect antibiotics because of their specific knowledge or because they may search for additional information via social media or newspapers. For example, they questioned whether it is necessary to finish the whole course of antibiotics because they noticed that scientific publications disagree about this matter. Furthermore, medium/higher-educated participants might have other motives to use antibiotics, because they mainly desired antibiotics due to upcoming social events.

Language-related barriers and feelings of being misunderstood are central problems in the healthcare of immigrants. This is not only in line with previous findings among patients [43], but is also underlined by GPs [38,44,45]. GPs emphasize the need for sufficient time during consultation and good professional interpreting services to overcome these problems [38,44], especially because the lack of time for explanation and shared decision-making increases antibiotic prescribing [46]. Additionally, when a GP perceives a demand of a patient for antibiotics, this significantly affects their prescribing [47]. This indicates that, patient-related, as well as healthcare professional-related, factors influence the inappropriate use of antibiotics. Therefore, it is important to have enough time for GP–patient communication, which helps build a trust-based relationship and mutual understanding. Immigrants are eager to learn and most of them want to receive information from their GP instead of receiving information via other channels, which is also stipulated in other studies [21]. GPs need to be aware of this, as they often withhold information because they expect that immigrants do not need, want, or understand this [48]. Our results support previous research that has shown that multi-language information materials are not always available. These materials are one of the first and most critical interventions to improve healthcare for immigrants because they can help to overcome language barriers [21,49]. Language barriers can cause patients to be misunderstood and, ultimately, be misdiagnosed [38]. In addition, language barriers can lead to wrong estimations of GPs about patients’ expectations. Various studies showed that GPs overestimate patients’ expectations for antibiotics and prescribe antibiotics to maintain a good patient–GP relationship [50].

Native populations can also have difficulties with understanding their GP and with following treatment advice, even if there is no language barrier, as we have shown in this study. Other quantitative studies among the native population also concluded that the public’s understanding about antibiotics needs to be improved [51,52]. To stimulate more prudent antibiotic use, it is necessary that GPs adjust their provided guidance and communication to individual needs, regardless of migration background, and that they refrain from stereotyping during the consultation by, for example, always asking about the expectations of the individual patient. Highly detailed and individualized explanations are supportive in getting across the decision not to prescribe antibiotics [46]. Within the context of reducing antimicrobial resistance, it is already stipulated that communication should be framed in a manner that does not enhance the high level of stigmatization of immigrants [53].

Our study had some limitations. First, we included a relatively small number of newly-arrived immigrants. However, we still succeeded in including a diverse immigrant group that is representative of the Dutch population because most Dutch immigrants have been living in the Netherlands for multiple years. Second, the immigrant group composition (mostly women, all age groups) was slightly different from that of the native Dutch group (men and women, mostly over 60 years). This slightly hampered the ability to make a full comparison and underlines the need to interpret our results with caution. Third, for some participants, translations in Dutch or English were required, which might have resulted in some loss of information. However, we tried to avoid this by using the Listening, Summarizing, and Digging deeper (LSD) method to ensure we understood everything correctly. Fourth, we realize that the background characteristics of the moderators (both white, female, highly educated) might have influenced the FGDs. In being aware of this, the moderators tried to ensure the participants felt as comfortable as possible, and that there was enough time to get familiarized during an extensive introduction, coffee breaks, and shared meals afterwards.

Future research on antibiotic perceptions could focus on what causes the decline in expectations to receive antibiotics and whether there is a similar trend among all population groups. Gaining knowledge about this will be helpful in shaping effective interventions, for all types of medication and treatments, and in deciding which population groups need to be targeted. Furthermore, attention is needed to improve the communication between GPs and their patients and for the development of patient information materials in multiple languages because they can help overcome language barriers and are necessary to inform and educate patients. In the PARCA-project, we used the results of the FGDs to develop a communication intervention for GPs [54].

## 11. Conclusions

There is considerable variation among individuals from the same population group regarding antibiotic knowledge, attitude, and information preferences. Yet, in general, having a reluctant antibiotic attitude is dominant. Instead of placing too much focus on someone’s background, it is necessary to adjust the provided information, guidance, and communication to individual needs in order to stimulate more prudent use of antibiotics. There should be specific attention given to patients who encounter language barriers because this is an impediment to the delivery of high-quality antibiotic management. Patient information materials in multiple languages should be developed and implemented on a broader scale.

## Figures and Tables

**Table 1 antibiotics-11-01179-t001:** Background characteristics of participants of the focus group discussions (N = 64).

	Participants with a Migrant Background						Native Dutch Participants
	Turkish N = 11	Moroccan N = 8	Surinamese N = 10	Cape Verdean N = 13	Syrian N = 4	Total (N = 46) (%)	N = 18 (%)
**Gender**							
Men		8	3			11 (24%)	9 (50%)
Women	11		7	13	4	35 (76%)	9 (50%)
**Age**							
Range	30–46	43–67	39–75	43–76	28–36	28–76	30–84
20–30	1				2	3 (7%)	1 (6%)
31–40	7		2		2	11 (24%)	2 (11%)
41–50	3	2	1	1		7 (15%)	2 (11%)
51–60		4	1	3		8 (17%)	2 (11%)
61–70		2	1	5		8 (17%)	8 (44%)
>70			5	4		9 (20%)	3 (17%)
**Educational level ^1^**							
Low	5	6	5	11	2	29 (63%)	10 (56%)
Medium	5	2	3	2	2	14 (30%)	
High	1		2			3 (7%)	8 (44%)
**Years living in the Netherlands (NL)**							
Not applicable (born in NL)	5					5 (11%)	18 (100%)
<10					4	4 (9%)	
10–20	3	3		1		7 (15%)	
21–30	1	1	1	1		4 (9%)	
31–40	2	3	2	5		12 (26%)	
>40		1	7	6		14 (30%)	

^1^ Low education: Persons whose highest level of education is primary education or vmbo (preparatory secondary vocational education). Medium education: Persons who have graduated from mbo (secondary vocational education), havo (higher general secondary education), or vwo (preparatory university education). High education: Persons who followed an associate degree program or higher education (HBO/WO).

## Data Availability

All transcripts used for the analysis are available from the corresponding author upon request.

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
