# Peer review of "Perspectives of Immigrants and Native Dutch on Antibiotic Use: A Qualitative Study"

_antibiotics, 2022, doi:10.3390/antibiotics11091179_

Round 1

Reviewer 1 Report

The presented work is fascinating and holds some potential, however, some comments are listed below to the authors for their response and considerations

1. in the abstract please add software to "Atlas.ti."

2. I presume that the audio recording was done individually please clarify

3. Authors should expand on their reasoning for expecting a difference between natives and immigrants in terms of their knowledge, specialists, and shared knowledge of antibiotics misuse. For example, most immigrants have to be highly qualified to be granted migration visas and most are GPs, engineers so they are highly educated and therefore should understand the importance of the correct use of antibiotics

4. was there any particular reasons for the selection of the immigrants from each of the countries or was it selected randomly based on the set recruitment procedure?

5. it is advisable for authors to include their approach to developing the questionnaire and should also include the local IRB agreement and whether the language was the immigrant's native language to ensure accurate data was collected from the recordings

6.  would it be possible to include in the table the participants or age range at least

7. what is defined by the authors as "lower education" what does this mean 

8. I believe there should be a part in which the questionnaire evaluates the level of knowledge about antibiotics (poor. moderate, good etc)

9.  was there any association of demographic characteristics with the level of knowledge of participants about antibiotics' correct use

10.  Knowledge and attitudes towards proper antibiotic usage, among  people regardless of where they come from should be highlighted within the manuscript

11. The public misuse of antibiotics is not the only determining factor for this issue as healthcare providers should be held accountable as well. Unnecessary prescriptions and easy accessibility to over-the-counter antibiotics, such as generic types of penicillin and cephalosporin, etc should be included in the discussion and highlighted as another influential factor

12.  would it be possible to use a demographic distribution of responses to the statements about antibiotics and present for example histograms showing the difference between the level of antibiotic knowledge with the questionnaire settings

13. was there any association between antibiotic use and attitude statements and awareness of dangers associated with the incorrect use of antibiotics

14. English language should be revised as many typos and grammatical mistakes are presented within the manuscript. I belive the manuscript would benefit from native speaker touch

15. several studies from Netherland looked into similar approaches their work at least should be cited for example PMID: 35501699; PMID: 31935856; PMID: 14519251 

Author Response

Response to Reviewer 1 Comments

Dear reviewer,

Thank you for your valuable comments and the opportunity to revise our manuscript. Please, find our detailed point-by-point response below. We are confident that these modifications have improved our manuscript.

  1. In the abstract please add software to “Atlas.ti”.
  • We changed this accordingly (lines 22-23).

  1. I presume that the audio recording was done individually please clarify.
  • We describe in the manuscript that all focus groups discussions (FGDs) were audio-recorded (line 163), meaning that the recordings of each separate FGD contains the answers of all participants. We did not make audio recordings of individual answers as a FGD is a research technique that collects data through group interaction and, as such, collects data on group level.

  1. Authors should expand on their reasoning for expecting a difference between natives and immigrants in terms of their knowledge, specialists, and shared knowledge of antibiotic misuse. For example, most immigrants have to be highly qualified to be granted migration visas and most are GPs, engineers so they are highly educated and therefore should understand the importance of the correct use of antibiotics.
  • Thank you for your valuable suggestion. It is correct that part of the immigrants is highly educated and must be highly qualified to be granted migration visas. However, this mainly relates to the most recent immigrants. In the Netherlands there are also many immigrants with lower qualifications, for instance the first-generation immigrants who arrived, from the 1960s onwards, as ‘unqualified guest workers’. Furthermore, a large part of immigrants are refugees, originating from one of the former Dutch colonies, or arrived because of family reunification. In many countries where immigrants came from, antibiotics are more easily prescribed, and, in some countries, they are even available over-the-counter. This shapes the perceptions of people about antibiotics. Therefore, we expected immigrants to hold other perceptions about antibiotic use. We improved our Introduction by expanding on our reasoning for expecting a difference between natives and immigrants in terms of their antibiotic perceptions (lines 80-86).

  1. Was there any particular reasons for the selection of the immigrants from each of the countries or was it selected randomly based on the set recruitment procedure?
  • Reasoning for the selection of specific immigrant groups is explained in the Method section (lines 116-120). Agency coordinators of various immigrant organizations assisted in the recruitment of suitable participants: first-generation immigrants, with different educational levels, who had used antibiotics at least once (lines 126-131).

  1. It is advisable for authors to include their approach to developing the questionnaire and should also include the local IRB agreement and whether the language was the immigrants’ native language to ensure accurate data was collected from the recordings.
  • We used a topic-guide that was based on literature and reviewed by health care professionals and health literacy experts (lines 145-149). We added information on how we ensured that the topic-guide questions were understandable for all participants, also for those with limited language abilities and/or limited literacy skills (lines 149-152).
  • The study, including the topic guide, was reviewed and approved by the Medical Ethics Review Committee at Erasmus MC (lines 183-185). If required by the editor, we can send the letter of the Committee stating their approval.
  • All FGDs were in Dutch with a topic guide that contained simple, short questions. Most of the recruited immigrants have been living in the Netherlands for multiple years and have learned Dutch, but there were some who were less proficient in Dutch. If a participant had difficulties with understanding the questions or expressing in Dutch, one of the other participants acted as translator. In the Syrian group, two of the participants sometimes felt more comfortable in speaking English, which was a shared language with the moderator. We added this information to Results (line 196-199). In the limitations we acknowledge that the translations might have resulted in some loss of information (lines 635-638).
  1. Would it be possible to include in the table the participants or age range at least.
  • We added the age range to Table 1.
  • It is not possible to recognize individuals in the data as we only gathered group data and no data on individual level.

  1. What is defined by the authors as “lower education” what does this mean.
  • We agree that this term needs clarification and we added this in the legend of Table 1.

  1. I believe there should be a part in which the questionnaire evaluates the level of knowledge about antibiotics (poor, moderate, good, etc).
  • For this qualitative study we only gathered group level-data by using a topic-guide. As a result, we did not always receive an answer of each individual participant to a specific question. This makes it difficult to make general statements. We can carefully conclude, also from the quotes, that there is high variation in knowledge about antibiotics: within all groups it varied from poor to good knowledge. We included this finding in the Results (lines 289).

  1. Was there any association of demographic characteristics with the level of knowledge of participants about antibiotics’ correct use.
  • The aim of the study was not to collect individual data, therefore, we are unable to analyze possible associations at individual level. During the FGDs we did not notice any clear patterns that could be related to demographic characteristics.

  1. Knowledge and attitudes towards proper antibiotic usage, among people regardless of where they come from should be highlighted within the manuscript.
  • In the manuscript we describe the wide variety in knowledge and attitudes among individuals within the same population groups, also by using quotes. At the beginning of the Results, we explain that we did not find noteworthy differences between immigrant and native Dutch participants for most of the themes (lines 215-223).

  1. The public misuse of antibiotics is not the only determining factor for this issue as healthcare providers should be held accountable as well. Unnecessary prescriptions and easy accessibility to over-the-counter antibiotics, such as generic types of penicillin and cephalosporing, etc should be included in the discussion and highlighted as another influential factor.
  • We certainly agree that healthcare providers should be held accountable as well and described this in the Introduction (lines 44-50; 86-89). In the discussion we have also paid attention to health care provider related factors (lines 594-599). We have made some modifications to highlight this further (lines 600-604).

  1. Would it be possible to use a demographic distribution of responses to the statements about antibiotics and present for example histograms showing the difference between the level of antibiotic knowledge with the questionnaire setting.
  • This is an interesting suggestion. However, this was an exploratory qualitative research that used FGDs. As a result, we did not gather individual based answers.

  1. Was there any association between antibiotic use and attitude statements and awareness of dangers associated with the incorrect use of antibiotics.
  • In general, it seemed that participants who were more aware about the risks related to antibiotic use, were also more reluctant of using antibiotics. However, because of the qualitative nature of this study, we are unable to examine this association at participant level. We, therefore, did not include any statements about possible associations in our manuscript.

  1. English language should be revised as many typos and grammatical mistakes are presented within the manuscript. I believe the manuscript would benefit from native speaker touch.
  • Thank you for alerting us. We now have our manuscript reviewed by a native speaker, who made multiple corrections throughout the manuscript.

  1. Several studies from Netherlands looked into similar approaches their work at least should be cited for example PMID: 35501699; PMID: 31935856; PMID: 14519251.
  • Thank you for sharing these studies. PMID: 35501699 was already included in our article (45). Although the other two articles discuss antibiotic use and persons’ knowledge and attitude towards antibiotic usage, they describe pre-professional students from Saudi Arabia and American inhabitants. Because of this different focus and the multitude of other more relevant studies in our manuscript, we have decided not to include these two.

Reviewer 2 Report

Dear Editor,

General Comments

Many thanks for inviting me to review this paper. This study investigated the antibiotic use perspective of patients from different nationality. I write my suggestions below.

I believe this study aligns with the scope of the journal. Antibiotics is a highly reputable academic journal and has a distinguished audience. And its’ audience deserve high-quality and exquisite publications.

The methods and study framework are quite solid however the novelty is the point that makes me think about this paper. These concerns should be overcome by the authors.

There are some minor limitations of the study. I believe this study is significant enough to publish Journal of Antibiotics.

Title

I believe the title is suitable and adheres with the content of the study.

Abstract

·         The information about methods section is rather short.

·         I think more information should be given in abstract about methods such as What was the criteria for allocated to any FCDs? How did the group discussion developed, what was the evaluation criteria, how analyses performed and interpreted etcetera?

·         The take home message is quite broad. I believe detailed outcomes should be underlined. I would like to suggest authors to give some details about the results and outcomes.

·         Keywords: I would like to recommend the adhere MeSH headings.

·         Please adhere the journal guideline especially for reference sections. Some references have DOI number some don’t, there are few reference without containing any page number etcetra.

Introduction

·         I believe the introduction section is well organized and beneficial.

Methods

·         In line 93-95 the authors were mentioned about the advantages of FCD? Is it really necessary to mention in methods section since the audience is experts about these kinds of investigations and already familiar with qualitative studies? On the other hand, each method has pros and cons. However only pros of FCDs are detailed within the text.

·         Since this study involved immigrants, in which language this FCD are conducted? I believe all FCDs were carried out in Dutch. Since the participants are first generation immigrants, I believe the language should be a barrier. If so, how did the researchers overcome this problem?

·         It would be useful to see the COREQ checklist as a supplementary table. The authors might be added such list, but it is not accessible for me.

·         It would be nice to see the level of income of the participants, since it would have led the bias.

Results

·         In line 333-335 and 339-342 the quotations should be written in italic.

Discussion

·         There is a typo mistake in line 559

·          

References

·         Need to adhere to the guidance for authors.

·         It would be better to give up to date references.

Author Response

Response to Reviewer 2 Comments

Dear reviewer,

Thank you for your valuable comments and the opportunity to revise our manuscript. Please, find our detailed point-by-point response below. We are confident that these modifications have improved our manuscript.

Title

I believe the title is suitable and adheres with the content of the study.

  • Thank you.

Abstract

  1. I think more information should be given in abstract about methods such as What was the criteria for allocated to any FCDs? How did the group discussion developed, what was the evaluation criteria, how analyses performed and interpreted etcetera?
  • Within the acceptable number of words, we have included more information about the methods in the Abstract now (lines 17-23).

  1. The take home message is quite broad. I believe detailed outcomes should be underlined. I would like to suggest authors to give some details about the results and outcomes.
  • We changed the take home message to make it more tangible (lines 34-36).

  1. Keywords: I would like to recommend to adhere MeSH headings.
  • We have changed the keywords into standardized MeSH terms (lines 39-40).

  1. Please adhere the journal guidelines especially for reference sections. Some references have DOI number some don’t, there are few reference without containing any page number etcetra.
  • We have carefully reviewed the References and added information where needed. It was not possible to add a DOI number (ref 47) and/or page number to all articles; articles that are only published online not always have page numbers.

Introduction

I believe the introduction is well organized and beneficial.

  • Thank you.

Methods

  1. In line 93-95 the authors were mentioned about the advantages of FCD? Is it really necessary to mention in methods section since the audience is experts about these kinds of investigations and already familiar with qualitative studies? On the other hand, each method has pros and cons. However only pros of FCDs are detailed within text.
  • Thank you for this valuable comment. We agree it is unusual to add advantages of the used method to the Methods. We, therefore, removed this sentence.

  1. Since this study involved immigrants, in which language this FCD are conducted? I believe all FCDs were carried out in Dutch. Since the participants are first generation immigrants. I believe the language should be a barrier. If so, how did the researchers overcome this problem?
  • All FGDs were in Dutch with a topic guide that contained simple, short questions. Most of the recruited immigrants have been living in the Netherlands for multiple years and have learned Dutch, but there were some who were less proficient in Dutch. If a participant had difficulties with understanding the questions or expressing in Dutch, one of the other participants acted as translator. In the Syrian group, two of the participants sometimes felt more comfortable in speaking English, which was a shared language with the moderator. We added this information to Results (line 196-199). In the limitations we acknowledge that the translations might have resulted in some loss of information (lines 635-638).

  1. It would be useful to see the COREQ-checklist as a supplementary table.
  • It would be possible to add the COREQ-checklist if the editor wants us to. If so, we need to elaborate a bit further on certain details in the manuscript that will extend the number of words. Please, let us know your preference.

  1. It would be nice to see the level of income of the participants, since it would have led the bias.
  • We agree this information would have been interesting. However, we did not collect these data. We decided to ask about the educational level, as a less sensitive topic.

Results

  1. In line 333-335 and 339-342 the quotations should be written in italic.
  • We changed this accordingly.

Discussion

  1. There is a typo mistake in line 559.
  • We changed this accordingly.

References

11.Need to adhere to the guidance for authors.

  • We have carefully reviewed the References and added information where needed. It was not possible to add a DOI number (ref 47) and/or page number to all articles; articles that are only published online not always have page numbers.

  1. It would be better to give up to date references.
  • We have carefully reviewed our references and feel we have relevant and current studies included. More than half of the references (57%) were published in the past five years. If there are suggestions for more recent publications, we are eager to hear about those.

Reviewer 3 Report

1. The sample size seems very low compared to the data claim of the entire Dutch land. Hence the title needs to be altered to correlate with the study.

2. Statement claim to be data of the entire country

3. Need to collect more opinions from the subjects about their understanding of antibiotic usage.

4. Transcribed version can be used after the significance of statistics

Author Response

Response to Reviewer 3 Comments

Dear reviewer,

Thank you for your valuable comments and the opportunity to revise our manuscript. Please, find our detailed point-by-point response below. We are confident that these modifications have improved our manuscript.

  1. The sample size seems very low compared to the data claim of the entire Dutch land. Hence the title needs to be altered to correlate with the study.
  • Thank you for your critical comment. We feel, we do not claim that we present data of the entire Netherlands. In the title we therefore refer to the focus of our study and stated ‘perspectives of immigrants and native Dutch…’. In the abstract we now added that the FGDs were held in Rotterdam and Utrecht (line 20).

  1. Statement claim to be data of the entire country
  • In the Methods we describe that we recruited a selection of immigrants and a selection of native Dutch inhabitants. To our knowledge, there is no claim that we have data of the entire country. For clarity, we added in the abstract that FGDs were held in Rotterdam and Utrecht (line 20).

  1. Need to collect more opinions from the subjects about their understanding of antibiotic usage.
  • This indeed would have been very interesting. We performed an exploratory study and state, at the end of the Discussion, that more research is needed. Especially immigration groups need to be involved in research to learn more about their understanding of antibiotic usage.

4.Transcribed version can be used after the significance of statistics.

  • We did not perform any quantitative statistical analyses. The aim and collected data were group-level data, which makes it impossible to perform statistical analyses.

Reviewer 4 Report

Line 104 add reference.

Author Response

Response to Reviewer 4 Comments

Dear reviewer,

Thank you for your valuable comment and the opportunity to revise our manuscript. Please, find our detailed point-by-point response below. We are confident that this modification have improved our manuscript.

  1. Line 104 add reference.
  • We added the reference (line 117).